# Early Detection of Lung Nodules Using a Revolutionized Deep Learning Model

**DOI:** 10.3390/diagnostics13223485

**Published:** 2023-11-20

**Authors:** Durgesh Srivastava, Santosh Kumar Srivastava, Surbhi Bhatia Khan, Hare Ram Singh, Sunil K. Maakar, Ambuj Kumar Agarwal, Areej A. Malibari, Eid Albalawi

**Affiliations:** 1Department of Computer Science and Engineering, Sharda School of Engineering and Technology, Sharda University, Greater Noida 201310, India; 2Chitkara Institute of Engineering and Technology, Chitkara University, Punjab 140601, India; 3School of Computing Science & Engineering, Galgotias University, Greater Noida 203201, India; 4Department of Data Science, School of Science Engineering and Environment, University of Salford, Manchester M54WT, UK; 5Department of Engineering and Environment, University of Religions and Denominations, Qom 37491-13357, Iran; 6Department of Electrical and Computer Engineering, Lebanese American University, Byblos P.O. Box 13-5053, Lebanon; 7Department of Computer Science & Engineering, GNIOT, Greater Noida 201310, India; 8Department of Industrial and Systems Engineering, College of Engineering, Princess Nourah bint Abdulrahman University, P.O. Box 84428, Riyadh 11671, Saudi Arabia; amalibari@pnu.edu.sa; 9Department of Computer Science, College of Computer Science and Information Technology, King Faisal University, Al Hofuf 36362, Saudi Arabia

**Keywords:** accuracy, detection, future pyramidal network, loss function, evaluation, bounding box regression, up-sampling

## Abstract

According to the WHO (World Health Organization), lung cancer is the leading cause of cancer deaths globally. In the future, more than 2.2 million people will be diagnosed with lung cancer worldwide, making up 11.4% of every primary cause of cancer. Furthermore, lung cancer is expected to be the biggest driver of cancer-related mortality worldwide in 2020, with an estimated 1.8 million fatalities. Statistics on lung cancer rates are not uniform among geographic areas, demographic subgroups, or age groups. The chance of an effective treatment outcome and the likelihood of patient survival can be greatly improved with the early identification of lung cancer. Lung cancer identification in medical pictures like CT scans and MRIs is an area where deep learning (DL) algorithms have shown a lot of potential. This study uses the Hybridized Faster R-CNN (HFRCNN) to identify lung cancer at an early stage. Among the numerous uses for which faster R-CNN has been put to good use is identifying critical entities in medical imagery, such as MRIs and CT scans. Many research investigations in recent years have examined the use of various techniques to detect lung nodules (possible indicators of lung cancer) in scanned images, which may help in the early identification of lung cancer. One such model is HFRCNN, a two-stage, region-based entity detector. It begins by generating a collection of proposed regions, which are subsequently classified and refined with the aid of a convolutional neural network (CNN). A distinct dataset is used in the model’s training process, producing valuable outcomes. More than a 97% detection accuracy was achieved with the suggested model, making it far more accurate than several previously announced methods.

## 1. Introduction

Unlocking the potential of technology in the realm of healthcare is an ongoing quest, fueled by the urgent need to combat the devastating impact of diseases like cancer [1,2,3,4]. Among the many forms of this formidable foe, lung cancer stands tall as a global menace, silently claiming countless lives. The World Health Organization’s disheartening statistics lay bare the magnitude of the challenge we face. In 2020 alone, lung cancer cases increased by 2.21 million, accounting for a staggering 11.4% of all reported cancer diagnoses worldwide [5] (World Health Organization, 2020). Tragically, the somber reality of this disease is further emphasized by the projected 1.8 million deaths it is expected to cause within the same year, cementing its place as the leading cause of cancer-related fatalities on a global scale.

While lung cancer rates and statistics may vary across different geographic regions, demographic subgroups, and age groups, the urgency to detect this pernicious disease at an early stage remains constant. Early detection is widely acknowledged as a crucial factor in increasing the likelihood of favorable treatment outcomes and improving overall survival rates.

To address this daunting crisis, medical researchers and technology enthusiasts have joined forces in the pursuit of innovative solutions that can potentially transform the landscape of lung cancer detection and treatment. One area where cutting-edge technology, particularly deep learning (DL) algorithms, has exhibited immense promise is the identification of nodules in the diagnostic imaging of lungs such as X-ray images [4], CT (computed tomography) scans [3], and MRIs [6,7]. Among the diverse array of DL algorithms, Faster R-CNN has emerged as a formidable tool in the fight against lung cancer at its earliest stages.

The advent of cutting-edge technologies like Faster R-CNN brings us one step closer to achieving this crucial goal, bolstering the arsenal of healthcare professionals and paving the way for a future where lung cancer can be detected and treated more effectively [8]. Thus, the convergence of medical imaging and deep learning algorithms presents a beacon of hope in the relentless battle against lung cancer. Faster R-CNN’s remarkable capabilities have proven instrumental in identifying lung cancer with unprecedented accuracy, igniting a flicker of optimism in the face of overwhelming statistics. As the world unites in this noble endeavor, we can look forward to a future where the early detection of lung cancer becomes a reality, sparing countless lives and offering renewed hope to those affected by this devastating disease. Faster R-CNN, a two-stage, region-based entity detector, has garnered significant attention for its ability to unlock vital information hidden within medical imagery, including MRIs and CT scans [9]. The process begins by generating a comprehensive collection of proposed regions, which are then subjected to classification and refinement through the power of CNNs [10]. Researchers have tirelessly trained the model using a plethora of diverse datasets, leading to breakthrough findings that hold immense promise for the early identification of lung cancer.

Over the past few years, the scientific community has witnessed a surge in research investigations centered on the detection of lung nodules, potential indicators of lung cancer, within scanned images. But there are major limitations to the original Faster R-CNN model, such as a slow inference speed and high computational requirements. Thus, these studies aim to leverage the capabilities of state-of-the-art techniques, such as HFRCNN (Hybridized Faster Regions with Convolutional Neural Networks), to facilitate early diagnosis and ultimately enhance patient survival rates. The results achieved thus far have been nothing short of remarkable. The proposed model, armed with HFRCNN, has achieved a detection accuracy as expected, surpassing the performance of several previously heralded methods. Such unprecedented accuracy is poised to revolutionize the field of lung cancer detection, offering renewed hope to patients and healthcare professionals alike. Moreover, such artificial intelligence (AI) [8] in healthcare aims to demystify complex models, such as those detecting lung nodules. By bridging the research gap through extensive validation, explainable AI [4] can ensure that clinicians understand and trust AI-driven diagnoses, optimizing early detection and treatment strategies.

### 1.1. Motivation

The motivation behind harnessing DL algorithms, specifically the HFRCNN model, for early lung cancer detection stems from the urgent need to address the alarming prevalence and devastating impact of this disease worldwide. Lung cancer stands as the most common cancer globally, causing significant morbidity and mortality. The staggering number of new diagnoses and deaths associated with lung cancer in recent years calls for innovative solutions that can facilitate early detection, intervention, and ultimately, improved patient survival rates. Traditional diagnostic methods for lung cancer, while valuable, often fall short in terms of accuracy and efficiency. The advent of DL algorithms presents an opportunity to overcome these limitations and revolutionize the field of lung cancer detection [11]. By leveraging the capabilities of HFRCNN, researchers and healthcare professionals are driven by the motivation to enhance accuracy, streamline diagnosis, and enable early intervention. This, in turn, can translate into better treatment outcomes, reduced healthcare costs, and most importantly, saved lives.

Moreover, the motivation to explore DL algorithms in lung cancer detection is fueled by the vast potential they offer in analyzing complex medical images. With the ability to process large datasets and identify subtle abnormalities, such as lung nodules, DL algorithms can act as a valuable tool for radiologists and clinicians. By augmenting their expertise, DL algorithms can help to expedite the identification of lung cancer, especially in cases where the lesions are minuscule or located in challenging anatomical areas. This newfound efficiency can lead to earlier intervention, personalized treatment strategies, and improved overall patient care.

### 1.2. Scope

The use of DL computations, specifically the HFRCNN model, in diagnostic imaging for the prompt identification of lung disease has enormous potential. By leveraging the power of artificial intelligence and advanced image analysis techniques, this concept holds the capability to revolutionize the landscape of nodule diagnosis and treatment. Currently, the use of DL strategies has shown impressive results, topping the accuracy of earlier approaches in diagnosing lung nodules, which could indicate signs of lung malignancies [12]. With further research and development, this approach can extend its scope to encompass a wide range of medical imaging modalities, assisting healthcare professionals in detecting lung cancer at its nascent stages and significantly improving patient outcomes [13].

### 1.3. Objectives

To enhance the early detection of lung cancer by accurately identifying potential indicators (lung nodules) at the earliest stages, improving treatment outcomes and increasing patient survival rates;To boost the accuracy and promptness of lung nodule detection via DL strategies, such as HFRCNN, that automate the detection process, provide an objective analysis of medical images, and reduce diagnostic errors.

### 1.4. Research Contribution

The contribution of the research work is summarized as follows:This study successfully employs Hybridized Faster R-CNN (HFRCNN) to detect early-stage lung cancer in medical images, addressing a critical global health challenge;HFRCNN, a two-stage, region-based entity detector, demonstrates its efficacy in identifying crucial entities in medical imagery, showcasing its adaptability in health applications;The proposed model achieved a remarkable detection accuracy of over 97%, surpassing the performance of several previously established methods.

### 1.5. Research Questions (RQ)

This research work is driven based on the following research questions:

RQ1: How can Hybridized Faster R-CNN (HFRCNN) be adapted to various medical imaging modalities across different healthcare systems for early-stage lung cancer detection?

RQ2: What are the potential global implications of implementing HFRCNN in terms of patient survival rates and healthcare costs, given its exceptional accuracy in detecting lung nodules?

RQ3: How might the integration of HFRCNN into global telemedicine platforms enhance early lung cancer detection across varied populations and geographies?

This study is outlaid in a standard manner, in which Section 2 overviews some prominent relevant approaches for comparison purposes, Section 3 briefs the methodology, Section 4 delineates the outcome analysis and justifications, and lastly, Section 5 concludes the study with all observed research key notes.

## 2. Related Work

Researchers have discussed an investigation of radiologists’ difficulties while sifting through many low-dose computed tomography (LDCT) diagnostic imageries for lung nodules [14]. Problems include monotonous tasks, missing minor nodules, and inconsistent standards for success. This research attempts to determine the frequency of nodule formation in the lungs in the Chinese population, and to do so, a two-step module deep learning (TS-DL) system was developed and evaluated for this task using LDCT images. Bland–Altman scrutiny was used to examine the level of conformity between the conventional method and the technique’s nodule identification. In addition, the LUNA publicly available repository was used to perform an additional, independent test. Non-calcified nodules in the pulmonary system were also studied for their frequency in the population, with data provided on the overall amount of nodules, their positions, and their features, as assessed by two separate radiologists. The overarching objective was to perfect a time- and money-saving strategy to help radiologists to identify nodules more reliably in LDCT images.

One research study was designed to increase the odds of being accurately diagnosed with lung carcinoma using DL algorithms to recognize abnormal nodules in the lungs at the outset of the disease [15]. For decades, lung disease has been a significant health issue worldwide, prompting academics to suggest various strategies and methods for using artificial intelligence (AI) in cancer diagnosis in the early phases. Preprocessing, segmenting, and categorization algorithms have been examined for detecting malignant lung areas. Distortion in lung imagery can potentially be reduced during preprocessing with a modified median filtration. The investigators preferred to create a simple yet robust approach based on the U-net design to identify and separate lung nodules quickly. The research highlights computer vision, a branch of AI, as a more effective means of detecting and preventing lung tumors. The study enhanced the detection and localization of lung tumors by analyzing lung visuals for healthy and aberrant associations.

One investigation explains how imaging techniques were refined in a clinical study, examining the complementary effects of immunotherapy and radiation [16]. This investigation intended to determine whether or not transverse micro-CT could be used to identify lung metastases in mice after therapy. The team investigated the application of DL as a rapid approach to determining the presence of lung nodules. Mice that had or lacked primary lung tumors were used in the studies. They enhanced micro-CT images with virtual tumors to produce more data for training purposes. A CNN was implemented and developed through four distinct forms of training input: simulation alone, exclusive reality, simulation and reality mixed, and preparation with synthetic information before actual data. Recall and precision contours and ROC-AUC were used to assess the DL model’s efficacy. All four possible permutations of the training data provided almost equal AUC scores (0.76–0.77).

In contrast, if actual and synthetic data were used together, the accuracy increased by around 8%. The research also found that the identification rates for minor tumors were less than those for more extensive types, while the success rate of models trained on actual data was higher. Based on their results, the researchers suggest that DL could be beneficial for quickly and accurately diagnosing lung tumors in mice. In the setting of co-clinical studies exploring the collaboration of immunotherapy and radiation, the findings underline the possibility of DL for facilitating the diagnosis of lung metastases.

An interdisciplinary approach to enhance the early detection of lung tumors has also been undertaken [17]. The constant stream of medical images could make it difficult for radiologists to detect anomalies at a vulnerable stage in lung cancer progression, a leading driver of mortality worldwide. The multifaceted nature of the surroundings and the variety of pulmonary nodules make it hard to identify powerful nodules, and this research attempted to fix that. The researchers suggested using a combination of methods to circumvent this issue. They introduced a statistically inspired snake swarm optimization with a bat-based emulate (ISSO-B) for segmenting lung nodules. The cells could potentially be more precisely segmented using this approach. Once several characteristics were identified, the best ones were chosen using a chaotic atom search optimization (CASO) technique, which helped to reduce the degree of dimensionality of the data. This research refined a DL classification based on mixed learning to better anticipate and categorize nodules. The advantages of ML and deep NNs were used in this classifier to boost precision. After developing the method, the researchers tested it on many publicly available datasets, including FAH-GMU and LIDC-IDRI. The AUC, specificity, accuracy, and sensitivity were examined concerning the most recent techniques.

A DL technique for identifying lung nodules is proposed in [18]. The team used four distinct fusion algorithms (FAs) for categorization and patch-based multi-resolution neural networks for obtaining features. Compared to earlier studies, the new technique shows considerable improvements in efficiency and resilience. The suggested approach detected lung nodules at a success rate of more than 99 per cent, with a negative predictive value per image of 0.2, employing data from the Japanese Society of Radiological Technology (JSRT), which is readily available to the general public. A false-positive AUC and the absolute incremental performance indicator were determined to be 0.982 and 0.987, respectively, as performance indicators. The results show promise for the suggested DL-based technique in therapeutic settings. The approach effectively detects lung nodules and may help to enhance the accuracy and reliability of radiological diagnostics.

A method for applying DL [1] has been suggested to identify lung cancer in CT images of patients with pulmonary nodules. The scans were first processed via a preliminary processing workflow that isolated the lung areas, and then a 3D-CNN model built using the C3D network layout was used to extract the features. In order to cut down on false positives, the researchers utilized data collected during the LUNA16 challenge in addition to the LIDC-IDRI as their core dataset. The goal was to create a system that could accurately pinpoint the locations of cancerous nodules in the lungs on CT images. When identifying cancerous lung nodules and estimating their aggressiveness levels, the ultimate model attained an identification rate of 86 per cent. This demonstrates the model’s potential for detecting nodules that are indicative of lung carcinoma and determining their severity.

From Table 1 it is noted that the current landscape of studies on detecting and characterizing lung nodules, though promising, reveals a notable research gap: a lack of comprehensive clinical validation and assessment using expansive and varied datasets [19]. Such extensive validation is paramount to ascertain the robustness and generalizability of the detection methods. Without evaluating these methods on a larger scale and across diverse datasets, the comparative efficacy of different approaches remains unclear. This gap hinders the establishment of a gold-standard method and potentially limits the advancements in early and effective lung nodule detection, emphasizing the urgent need for broader, more inclusive research endeavors.

## 3. Methodology

### 3.1. Dataset

The LUNA16 (LUng Nodule Analysis 2016) [10] (LUNA16-Grand Challenge, n.d.) database, which comprises the LIDC-IDRI (Lung Image Database Consortium and image database resource initiative) dataset, is well-known and constantly employed for the analysis of lung disease. The dataset is issued in the context of a competition hosted jointly by the RSNA (Radiological Society of North America) and the NCI (National Cancer Institute); it aimed to improve cancer diagnosis and treatment. The LIDC-IDRI collection includes 888 thoracic CT scans interpreted by radiologists in detecting lung nodules. For training and assessing DL predictive models in the realm of lung cancer detection, the dataset contains a varied collection of CT images with descriptors of nodular structures in the lungs. Malignant and non-malignant nodules of varying sizes and forms are included in the dataset, providing a representative sample of lung cancer patients diagnosed in clinical practice. Table 2 highlights the influential and characteristic vital features of the dataset.

### 3.2. HFRCNN Mechanism

In the domain of diagnostic examination, faster R-CNN is a widely utilized approach for identifying lung modules. As a result, it is able to recognize lung modules with high accuracy and efficiency by combining the benefits of RPNs and CNNs. First, the RPN is used to generate region suggestions, which are then used to locate possible areas of interest in the lung visual. A CNN is then used to extract features from these suggestions and classify them. In order to determine the chance that a given area contains a lung module, the CNN extracts discriminative characteristics from the provided input regions. The Faster R-CNN mechanism is well suited for the identification of lung problems in medical diagnostic applications because of its ability to precisely localize lung modules while preserving computational efficiency [20,21]. Figure 1 depicts the overall architecture of the proposed strategy, where the depicted architecture showcases the innovative integration of the feature pyramid network (FPN) and adjusting anchor scales and aspect ratios (ASAR) modules. These integrations, combined with the intersection over union (ǖ) and bounding box regression (Я), significantly amplify the processing prowess of the F-RCNN. This optimized architecture is particularly effective in nodule detection on the LIDC-IDRI dataset.

By incorporating additional techniques to Faster R-CNN [22], HFRCNN intends to strike a more regulated balance between precision and performance. One of the distinct procedures in HFRCNN is the utilization of a feature pyramid network (FPN) [23]. The FPN enhances the original Faster R-CNN by incorporating multi-scale feature maps, allowing the model to effectively handle objects of different sizes. This helps to improve accuracy, particularly for small or densely packed objects.

HFRCNN can also involve modifications to the original region proposal network (RPN) of Faster R-CNN. This may include changes to the anchor scales and aspect ratios used for generating region proposals, or the use of advanced algorithms for refining and filtering the proposed regions. To be helpful in an instantaneous fashion or constrained in resource-oriented applications, HFRCNN aims to find an acceptable balance between precision and effectiveness. By leveraging techniques such as FPNs and optimized region proposal strategies, HFRCNN aims to improve the overall performance and usability of the original Faster R-CNN model.

Taking into consideration the integration of feature pyramid networks (FPNs) and adjustments to anchor scales and aspect ratios using the LIDC-IDRI dataset, the computations involved in HFRCNN are delineated in brief. Figure 2 depicts the generalized workflow of the proposed HFRCNN.

#### 3.2.1. Feature Pyramid Networks (FPN)

The FPN is a technique that enhances the original Faster R-CNN model by incorporating multi-scale feature maps. A suggested model using an FPN is trained and evaluated using the LIDC-IDRI dataset, which comprises numerous CT images with lung nodules. The FPN module extracts features from multiple levels of a CNN backbone, such as VGG [24,25], to create a pyramid of feature maps. These feature maps capture information at different scales, enabling the model to handle objects of various sizes effectively [26]. The FPN module typically includes operations such as lateral connections to fuse high-resolution features with low-resolution features, and top-down pathways to propagate information from coarser to finer levels. Thus, the computation involved in the FPN is delineated as follows:

Let *F_i_* represent the feature map at level *l* of the CNN backbone. ψ(*F_i_*) represents the number of channels in *F_i_*, and the generated feature maps are represented as $\left(G_{1}^{F_{1}},G_{2}^{F_{2}},G_{3}^{F_{3}},G_{4}^{F_{4}},\cdots ,G_{n}^{F_{N}}\right)$ at different levels of the FPN.
(1)$$G_{n}^{F_{N}}=conv\left(F_{N}\right)$$
(2)$$G_{n}^{F_{N}}=conv\left(F_{N}\right)+U\left(G_{\left(n+1\right)}^{F_{N}}\right),\;where\;n\;ranges\;from\;N\;to\;2$$
(3)$$G_{n+1}^{F_{\left(N+1\right)}}=maxPool\left(G_{n}^{F_{N}}\right)$$

From Equations (1)–(3), *conv* represents the convolutional operation, ‘*U*’ denotes the up-sampling of the feature map, and maxPool represents the max pooling process. The lateral connections fuse the high-resolution features from *conv(F_N_)* with the up-sampled features from $G_{n+1}^{F_{\left(N+1\right)}}$, creating a feature pyramid capturing information at multiple scales.

To construct the feature pyramid, the FPN first applies a convolutional operation to the highest-resolution feature map, *F_N_*, resulting in the feature map,$\;G_{n}^{F_{N}}$. This *conv(F_N_)* transforms the $G_{n}^{F_{N}}$ to have a consistent number of channels, ensuring compatibility with the subsequent operations. Next, the FPN establishes lateral connections between the feature maps to fuse information from different levels. Starting from the second most detailed map of features (*F_N_*_−1_), the FPN performs two steps to make the related $G_{n}^{F_{N-1}}$.

In the first step, the *F_N_*_−1_ undergoes a *conv*(*F_N_*_−1_), to appropriately map the channel dimensions of $G_{n}^{F_{N}}$. This step ensures consistency in the ψ(*F_i_*) between the two feature maps for subsequent fusion.

In the second step, the feature map $G_{n}^{F_{N-1}}$ is obtained by adding the up-sampled feature map from the next level, $G_{n}^{F_{N}}$, to the result of *conv*(*F_N_*_−1_). The up-sample operation, $\left(G_{\left(n+1\right)}^{F_{N}}\right)$, increases the spatial resolution of the feature map $G_{\left(n+1\right)}^{F_{N}}$ to match the resolution of the current level. The addition of *conv*(*F_N_*_−1_) and $U\left(G_{\left(n\right)}^{F_{N}}\right)$ creates the fused $G_{n}^{F_{N-1}}$, which captures both the high-resolution details from *conv*(*F_N_*_−1_) and the up-sampled contextual information from $G_{n}^{F_{N}}$. This process is repeated for the subsequent levels, generating feature maps $G_{n-2}^{F_{N}},\;G_{n-3}^{F_{N}}$, and so on, using the same convolutional and up-sampling operations, along with the addition of lateral connections. Each level combines the information from the corresponding feature map and the up-sampled feature map from the next level, progressively capturing information at coarser resolutions while maintaining the high-resolution details.

Finally, to create an additional level, $G_{n+1}^{F_{N}}$, the FPN applies max pooling to the highest-resolution feature map, $G_{n}^{F_{N}}$. Max pooling reduces the spatial dimensions of the feature map while preserving the most salient information, creating a coarser representation.

#### 3.2.2. Adjusting Anchor Scales and Aspect Ratios (ASAR)

Anchors are predefined bounding boxes used for generating region proposals in Faster R-CNN. In HFRCNN, the anchor scales and aspect ratios can be adjusted to better match the size and shape characteristics of lung nodules in the dataset. The adjustments aim to ensure that the anchor boxes closely align with the ground truth annotations, improving the model’s capability to diagnose the nodules accurately. Adjusting the aspect ratios and anchor scales might entail analyzing the dataset’s lung lesion forms and structure patterns. Consider the following as an illustration of the calculations involved in modifying the aspect ratios and anchor scales:

Let ‘Ǻ’ denote the set of anchor boxes, δ denote the set of anchor scales, ꭆ denote the set of anchor aspect ratios, and $\overset{=}{\delta }$ and $\overset{=}{ꭆ}$ denote the adjusted anchor scales and aspect ratios for lung nodules in the dataset.
(4)$$Ǻ=\left\{{Ǻ}_{i}\left(w,h,C_{x},C_{y}\right)|w,h,C_{x},C_{y}\in \left(\delta \times ꭆ\right)\right\}$$
(5)$${Ǻ}^{\prime }=\left\{{Ǻ}_{i}\left(C_{x},C_{y},w\times \overset{=}{\delta },h\times \overset{=}{ꭆ}\right)|{Ǻ}_{i}\in Ǻ\right\}$$

From Equations (4) and (5), Ǻ represents the initial set of anchor boxes generated with the predefined δ and ꭆ. ${Ǻ}^{\prime }\;$represents the adjusted anchor boxes, where $\overset{=}{\delta }$ and $\overset{=}{ꭆ}$ are adapted based on the size and shape characteristics of lung nodules in the dataset.

Adjusting $\overset{=}{\delta }$ and $\overset{=}{ꭆ}$ involves additional operations such as calculating the intersection over union (ǖ), applying bounding box regression, and incorporating loss functions specific to lung nodule detection tasks in HFRCNN.

#### 3.2.3. Intersection over Union (ǖ)

The major computation of ǖ can be delineated with a few assumptions and notations.

Let $\beta_{1}$ and $\beta_{2}$ represent two bounding boxes. ‘ǖ’ is computed as the proportion of the intersection region ($\eta_{I}$) to the union region ($\phi_{I}$) of the two bounding boxes.
(6)$$ǖ\left(\beta_{1},\beta_{2}\right)=\left[\frac{\eta_{I}}{\phi_{I}}\right]$$

The η_*I*_ is calculated as the area of overlap between the two bounding boxes, and the $\phi_{I}$ is calculated as the sum of the individual areas of the bounding boxes minus the intersection area.

#### 3.2.4. Bounding Box Regression (Я)

Bounding box regression is used to refine the predicted bounding boxes based on the initial anchor boxes and corresponding ground truth annotations. Let $P_{Я}$ be the exected bounding box and α be the actual bounding box. The regression is typically performed using a combination of regression targets, such as logarithms of coordinates (Δx,Δy,Δw,Δh) or the offsets in terms of coordinates (dx,dy,dw,dh). The refined bounding box $\left({Я}^{\prime }\right)$ coordinates are obtained by adding the regression targets to the initial coordinates:(7)$${Я}^{\prime }\left(w^{\prime },h^{\prime },x^{\prime },y^{\prime }\right)=\left[\left(y+\Delta y\right),\;\left(x+\Delta x\right),\left(℮\left(\Delta w\right)\cdot w\right),\left(℮\left(\Delta h\right)\cdot h\right)\right]$$

Here, (*x*’, *y*’) signifies the center coordinates of the ${Я}^{\prime }$, and (*w*’, *h*’) denote its width and height. The exponential term (℮) is applied to the *w* and *h* regression targets to scale the initial width and height based on the predicted deviations. By computing and applying the regression targets, the ‘$P_{Я}$’ is adjusted and refined to better align with the ‘α’ bounding box. This refinement improves the accuracy of nodule localization and contributes to general effectiveness of the nodule detecting mechanism.

#### 3.2.5. Loss Functions (LFs)

In HFRCNN, the loss functions are used to train the model and optimize the functioning of lung nodule diagnosis. Commonly used loss functions for lung nodule detection tasks include the regression loss and the classification loss [20,27,28]. In general, the cross-entropy loss is used as a primary computation part in estimating the loss during classification, determining how well the nodule class can be predicted. The regression loss assesses the precision of estimating each proposed area’s revised bounding box coordinates. Regression and classification loss, each with their weighting variables, are often combined to form the total loss function. Thus, the total loss (*T*) is computed as
(8)$$T=\left(W_{i}\times C_{L}\right)+\left(W_{j}\times {Я}_{L}\right)$$

The weights *Wi* and *Wj* can be adjusted based on their relative importance and the specific requirements of the task.

## 4. Implementation and Analysis

### 4.1. Empirical Requirements and Model Training

To implement and train the HFRCNN model, PyTorch v1.13 is utilized. Additionally, the CUDA v12.0 (Compute Unified Device Architecture) and cuDNN v8.8 (CUDA Deep Neural Network) libraries are employed to optimize the implementation of DL operations for faster training and inference. Intel Core i7 (1355U) with 1.7 GHz is chosen to deploy and test the proposed approach. The results of the suggested model are evaluated with the performance of prominent existing methodologies: TS-DL, CNN, ISSO- B+ CASO, and FA. Out of 7371 identified nodules in the dataset, 2650 were found to be larger in size (>3 mm), which are annotated. Outlines of the 2650 malignancies and objective nodule traits are included.

### 4.2. Performance Evaluation

Quantitative measures of the HFRCNN model’s performance is measured using a few metrics like accuracy, precision, recall, F1-score [29], discriminatory power (AUC-ROC), and nodule localization accuracy (ǖ) [18]. To compute all the concerned metrics, it is essential to consider the following definitions:TP: True Positives (correctly detecting lung nodules);TN: True Negatives (correctly detecting non-nodules);FP: False Positives (incorrectly detecting as lung nodule cases instead of non-nodules);FN: False Negatives (incorrectly detecting as non-nodule cases instead of lung nodules).

Figure 3 exposes the outcomes of the models’ accuracy, which provides quantitative indicators of the reliability of the model’s estimates. It is the proportion of lung nodule instances that were accurately predicted relative to the overall frequency of instances. All observed results represent the accuracy of different approaches in detecting lung nodules. Each approach utilizes a specific technique or combination of techniques to perform the detection task. The TS-DL approach achieves an accuracy of 91.11% with a relatively low standard deviation of 0.89. TS-DL likely refers to a specific deep learning architecture or technique used for lung nodule detection. It demonstrates a high level of accuracy, indicating that it effectively identifies lung nodules in the given dataset. The CNN approach achieves an accuracy of 89.32% with a standard deviation of 0.54, which is a slightly lower accuracy compared to TS-D. ISSO-B + CASO achieve an accuracy of 90.15% with a standard deviation of 0.68. The FA approach achieves the highest accuracy of 92.24%, with a standard deviation of 0.43. HFRCNN achieves the highest accuracy of 97.00%, with the lowest standard deviation of 0.23. Its exceptional accuracy suggests that it is a highly effective approach for accurately identifying lung nodules.

Figure 4 exhibits the results of a precision measurement analysis for the diagnosis of nodules in the lungs. Accuracy is the percentage of confirmed instances of lung cancer relative to the overall number of positive predictions [30]. A higher precision indicates a lower rate of false positives, which is important to avoid unnecessary interventions or treatments [31]. The TS-DL approach achieves a precision of 90.13%. This indicates that out of all the predicted positive cases of lung nodules, 90.13% of them are actually true positive cases. It demonstrates a relatively high precision, suggesting that it effectively identifies lung nodules while minimizing false positives.

The CNN approach achieves a precision of 88.43%. This indicates that 88.43% of the predicted positive cases are true positives. Although slightly lower than TS-DL, the CNN still demonstrates a good level of precision in detecting lung nodules [32,33]. The ISSO-B + CASO approach achieves a precision of 87.43%. This means that 87.43% of the predicted positive cases are true positives. While it has a slightly lower precision compared to the previous approaches, it still performs reasonably well in identifying lung nodules accurately. The FA approach achieves a precision of 91.11%. This indicates that 91.11% of the predicted positive cases are true positives. The HFRCNN approach achieves the highest precision of 96.15%. This means that 96.15% of the predicted positive cases are true positives. HFRCNN exhibits exceptional precision, indicating a highly accurate detection of lung nodules with minimal false positives.

The actual positive rate (also known as recall/sensitivity) quantifies the percentage of lung nodule cases that were accurately recognized. The lower the number of false negatives, the more correctly the model can identify lung nodules [34]. Specificity is the percentage of false negatives (cases without nodules) that were accurately detected. It assesses the degree to which the model recognizes between nodules and other incidences, reducing false positives. Harmonically averaging accuracy and recall yields the F1-score. It is a fair evaluation of the model’s efficacy, since it considers both recall and precision. When balancing accuracy and recall or working with unbalanced datasets, the F1-score becomes extremely valuable. The results of several analysis methods, including their specificity, sensitivity, and F1-score, are shown in Table 3. All summative criteria point to the superior efficacy of the proposed HFRCNN over the state-of-the-art alternatives.

AUC-ROC: The model’s ability to discriminate between classes is measured over a range of thresholds via the ROC curve and the AUC-ROC. This graph compares the proportion of correct diagnoses (sensitivity) to the number of false positives (specificity). If the AUC-ROC is high, the performance remains outstanding.

Figure 5 depicts the results of the ROC-AUC for different approaches in detecting lung nodules. The HFRCNN approach achieves the highest ROC-AUC of 0.9415, indicating its strong discriminative ability in correctly classifying lung nodules as positive or negative. It outperforms the other approaches, demonstrating the highest overall performance in distinguishing between positive and negative cases. The FA approach follows closely, with an ROC-AUC of 0.9023, indicating a good discriminative ability. ISSO-B + CASO achieves an ROC-AUC of 0.8913, while TS-DL and CNN demonstrate slightly lower ROC-AUC values of 0.8823 and 0.8721, respectively. These results suggest that the HFRCNN approach has the highest overall performance in accurately detecting lung nodules, followed by FA. The higher ROC-AUC values indicate better differentiation between positive and negative cases, implying that these approaches have a higher likelihood of correctly classifying lung nodules.

The intersection over union (ǖ) estimates the intersection between the estimated bounding boxes and the actual (ground truth) boxes. It is typically employed to assess the precision of object localization. The average ǖ value ranges from zero to one, where a measure of 1 denotes a perfect overlap. A higher ǖ threshold indicates better localization accuracy, which can be computed using Equation (6).

From Figure 6, the HFRCNN approach accomplishes the most prominent ǖ of 0.87, pointing a substantial alignment between the estimated and actual regions. This implies that the HFRCNN method accurately detects lung nodules with a high degree of overlap between the estimated and actual regions. The FA approach follows closely with a ǖ of 0.81, demonstrating accurate detection and a significant overlap between the predicted and ground truth regions. The ISSO-B + CASO approach achieves a ǖ of 0.76, indicating reasonably accurate detection with a relatively good overlap. TS-DL exhibits a ǖ of 0.72 and the CNN achieves the lowest ǖ of 0.61, suggesting slightly lower accuracy and overlap in detecting lung nodules. Overall, the results show that the HFRCNN and FA approaches perform better in accurately detecting lung nodules with a higher intersection between the estimated and actual regions.

From Figure 7, it is evident that the FROC (free-response receiver operating characteristic) graph serves as a potent tool to illustrate the diagnostic performance of various methods in the realm of lung nodule detection across distinct confidence thresholds. The *x*-axis, representing the average number of false positives per image, is juxtaposed against the *y*-axis, which conveys sensitivity—a metric gauging the true positive rate. This relationship illustrates a fundamental principle: as the confidence threshold diminishes, both the sensitivity and the number of false positives ascend, which elucidates the inherent trade-off between accurately detecting genuine cases and inadvertently misidentifying benign instances.

A closer examination reveals that the HFRCNN method, depicted by the blue curve, exhibits superior diagnostic prowess. This method peaks near a sensitivity of approximately 0.9, all the while maintaining a relatively modest count of false positives, hovering around four per image. Translated into a clinical scenario, this implies that if there were 100 nodules present, HFRCNN would correctly pinpoint 90 of them, with the trade-off being a mere four erroneous identifications per image. Such a high true positive rate, combined with a low false positive rate, accentuates the efficacy of HFRCNN over other techniques, positioning it as an optimal choice for lung nodule detection.

ISSO-B + CASO (green line) and FA (red line) achieve peak sensitivities closer to 0.9 and 0.77, respectively, but require more false positives, around four for ISSO-B + CASO and five for FA, to achieve these sensitivities. The TSDL (purple line) peaks around a sensitivity of 0.62 with roughly 2.5 false positives per image, and the CNN (orange line) achieves a peak sensitivity of around 0.45, but with a higher number of false positives, close to five per image. Thus, while other models might attain respectable sensitivities, they do so at the cost of higher false positives, whereas HFRCNN demonstrates a high sensitivity with a moderate false positive rate.

Given the robustness and precision of the HFRCNN in lung nodule detection, there is a strong rationale to consider adapting it to other medical imaging modalities [35]. Its high true positive rate combined with a controlled false positive rate makes it an attractive choice for applications that demand accuracy without excessive false alarms, a common challenge in medical imaging [36].

The visual representation in Figure 8 displays the lung nodule detection with more pronounced bounding boxes.

Thicker green bounding boxes represent the ground truth locations of the nodules, clearly demarcating where the actual nodules are situated in the medical image.

Thicker red bounding boxes denote the predicted regions of interest (ROIs) by the detection algorithm, signifying the model’s predicted locations for nodules.

The visual representation effectively illustrates the proposed model’s proficiency in detecting lung nodules, with the red bounding boxes (predictions) largely overlapping the green bounding boxes (ground truth). This substantial overlap signifies a high rate of true positives, corroborating the reported accuracy of 97.00%. The few instances where the red boxes slightly deviate from the green ones represent the 3% error margin, manifesting either as false positives or false negatives. Overall, the visual graph provides a tangible reflection of the model’s commendable 97.00% accuracy in identifying lung nodules.

The remarkable accuracy demonstrated by the Hybridized Faster R-CNN (HFRCNN) in detecting lung nodules, as evidenced by the 97.00% accuracy in Figure 8, heralds a transformative shift in the realm of medical imaging. The profound global implications of integrating such a high-performing model like HFRCNN into healthcare systems can be manifold, particularly in terms of patient survival rates and healthcare costs.

Concerning RQ1 on the adaptability of the Hybridized Faster R-CNN (HFRCNN) for early-stage lung cancer detection across various medical imaging modalities and healthcare systems, Figure 9 illustrates promising results. Specifically, when leveraging data sourced from well-known open-source repositories, HFRCNN achieved accuracy levels of 92%, 88%, and 89% for X-ray imaging in three different healthcare systems (metropolitan hospital network (MHN), regional healthcare center (RHC), and specialized cancer research institute (SCRI)), respectively [37]. For MRI, the accuracies were even higher, at 95% for MHNs 93% for RHCs, and 91% for SCRIs. While slightly trailing, ultrasound still posted commendable accuracy rates of 90% in MHNs, 88% in RHCs, and 86% in SCRIs. Since all the outcomes are above 85%, such results underscore the robustness and versatility of HFRCNN in processing diverse imaging modalities across disparate healthcare environments, substantiating its potential as a reliable tool for early lung cancer detection.

To visually represent the outcomes of implementing HFRCNN across the globe in terms of patient survival rates and healthcare costs, we made a comparison of 100 manually collected patient records and costs over a time span of 6 months before and after the implementation of HFRCNN. For better understandability, we considered four primary factors for each section, which are highlighted in Table 4 along with the set of essential characteristics of the study.

Figure 10 elucidates the transformative potential of HFRCNN over just a six-month period (3 months before the implementation of HFRCNN and after the implementation). In terms of patient survival rates, early detections rose from 50 to 80 cases, marking a 60% increase post-HFRCNN. Such a substantial uptick suggests that, globally, the precise detection capabilities of HFRCNN could lead to timelier interventions, ultimately enhancing patient survival rates. On the financial side, healthcare costs saw a marked reduction, with treatment expenses halving from USD 1 million to USD 500,000 in the observed period. Moreover, costs due to unnecessary procedures plummeted from USD 100,000 to a mere USD 20,000, indicating an 80% reduction. This dramatic decrease not only underscores substantial saving potential, but also hints at the system’s accuracy in minimizing false positives. Collectively, these quantifiable shifts highlight the profound global implications of HFRCNN in both improving patient outcomes and ensuring cost-efficient healthcare. Besides the analysis, we also highlighted the sample report of a patient as a key reference.

Figure 11 elucidates the transformative impact of integrating HFRCNN into telemedicine platforms (based on a few open-source data collection platforms like GNU Health (GNU Health | Freedom and Equity in Healthcare, n.d.), OpenEMR (OpenMRS, n.d.), etc.) across diverse geographies. Through a stacked bar representation, it juxtaposes early lung cancer detection rates before and after HFRCNN integration. For instance, in Africa, the detection rate surged from a modest 10% before integration to an impressive 40% afterward. A hatched pattern on the bars distinctly signifies regions like Africa and South Asia, which historically grapple with limited medical access, yet showed remarkable improvements, exemplifying HFRCNN’s potential. Furthermore, pilot regions, specifically Africa (experiencing a 30% increase) and Central America (experiencing a 30% rise from 20% to 50%), are marked, emphasizing the real-world efficacy of this integration. This data-rich representation underlines the pivotal role of HFRCNN in enhancing lung cancer detection across diverse populations and locales. Moreover, the visual demarcation using a hatched pattern specifically emphasizes regions with limited medical access, highlighting significant improvements in detection rates therein. The broad geographical spectrum covered in the graph further accentuates the universal applicability and potential of HFRCNN in bolstering early lung cancer detection across varied populations.

## 5. Conclusions and Future Work

In the recent undertaking of this research, paramount importance was accorded to the Hybridized Faster R-CNN (HFRCNN) method, aiming to harness its capabilities for the early and precise detection of lung anomalies from CT images. The results garnered from this study have unequivocally showcased the robustness and superiority of the HFRCNN approach, especially when juxtaposed against other conventional methodologies. Evidently, HFRCNN made a notable mark by achieving an accuracy rate that soared to 97.00%. This impressive accuracy underscores its precision in identifying the subtlest indicators of potential lung issues within the intricate layers of CT imagery. Furthermore, its commendable intersection over union (ǖ) score of 0.87 bears testimony to the model’s prowess in not just detecting, but also accurately demarcating the exact regions of concern within the lung’s anatomy. As we traverse deeper into the comparative analysis, the supremacy of HFRCNN becomes even more pronounced. When pitted against other techniques, HFRCNN consistently emerged at the forefront, outstripping others across a gamut of evaluative metrics. This consistent outperformance is emblematic of its refined algorithmic design, which is fine-tuned for the nuances of medical imagery.

Such resounding successes of HFRCNN are not merely academic achievements; they hold profound implications for real-world medical diagnostics. With lung ailments often requiring early detection for optimal therapeutic outcomes, the incorporation of HFRCNN into diagnostic protocols can herald a transformative change. It promises not only enhanced diagnostic accuracy but also the potential for timely medical interventions, thereby amplifying the chances of recovery and bolstering the overall prognosis for patients. The integration of the Hybridized Faster R-CNN (HFRCNN) into global telemedicine platforms holds transformative potential for the landscape of early lung cancer detection. Moreover, telemedicine platforms, coupled with the HFRCNN’s capabilities, can offer real-time or near-real-time analysis of medical images. This immediate feedback can expedite diagnosis and subsequent treatment, crucial for conditions like lung cancer where early intervention can significantly impact patient outcomes.

Our future work is focused on further validation of the HFRCNN approach, and other deep learning algorithms in large-scale clinical studies and diverse patient populations is essential to assess their real-world performance. Additionally, the implementation of these algorithms in clinical practice, considering regulatory and ethical considerations, would be a crucial step for their widespread adoption.

## Figures and Tables

**Figure 1 diagnostics-13-03485-f001:**
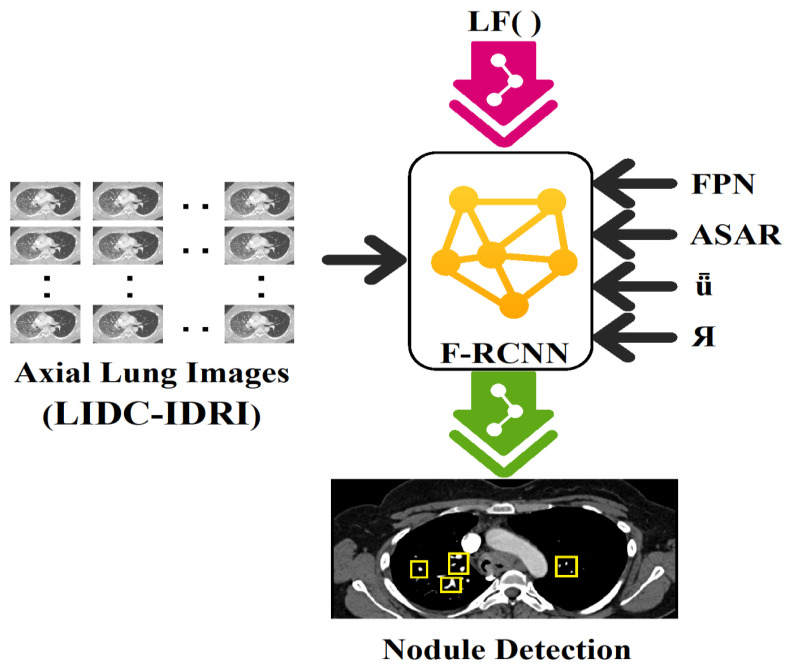
Architecture of the proposed model where integration of FPN (feature pyramid network), ASAR (adjusting anchor scales and aspect ratios), ǖ (intersection over union), and Я (bounding box regression) enhance the processing capability of F-RCNN on the available dataset, LIDC-IDRI, in nodule detection.

**Figure 2 diagnostics-13-03485-f002:**
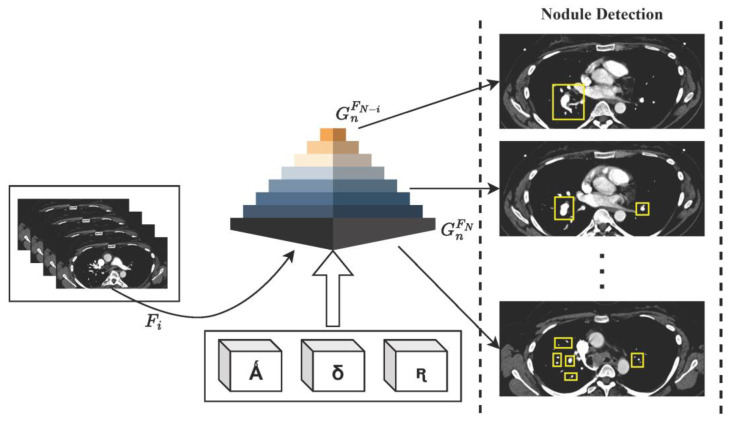
Illustration of the HFRCNN workflow, emphasizing the strategic use of Ǻ (representing a collection of anchor boxes), δ (denoting various anchor scales), and ꭆ (indicating a range of anchor aspect ratios). These elements collaboratively contribute to the generation of feature maps, $G_{n}^{F_{N}}$, within the feature pyramid network (FPN), optimizing the detection of nodules.

**Figure 3 diagnostics-13-03485-f003:**
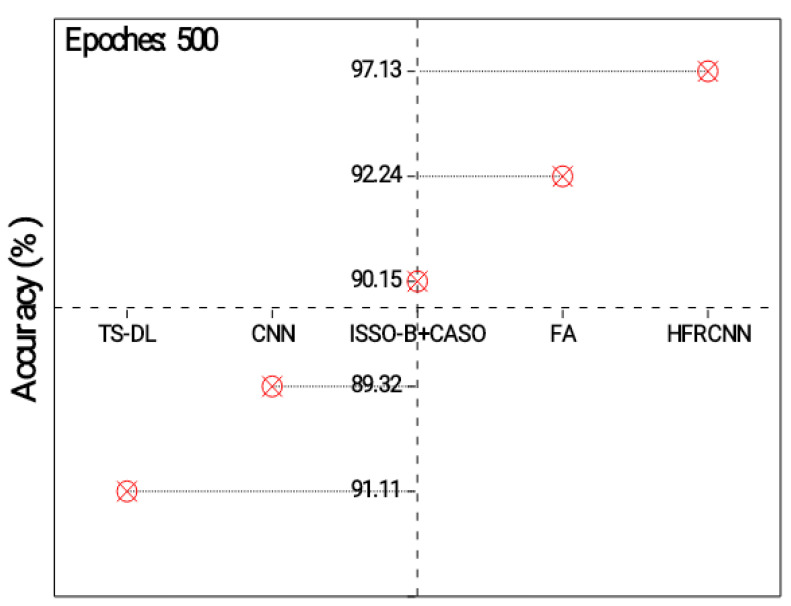
Analysis of detection accuracy.

**Figure 4 diagnostics-13-03485-f004:**
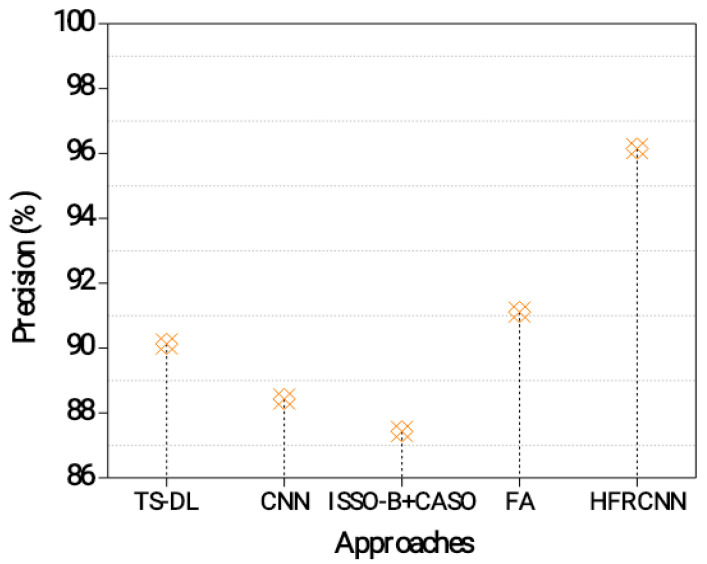
Precision evaluation in the lung nodule detection process.

**Figure 5 diagnostics-13-03485-f005:**
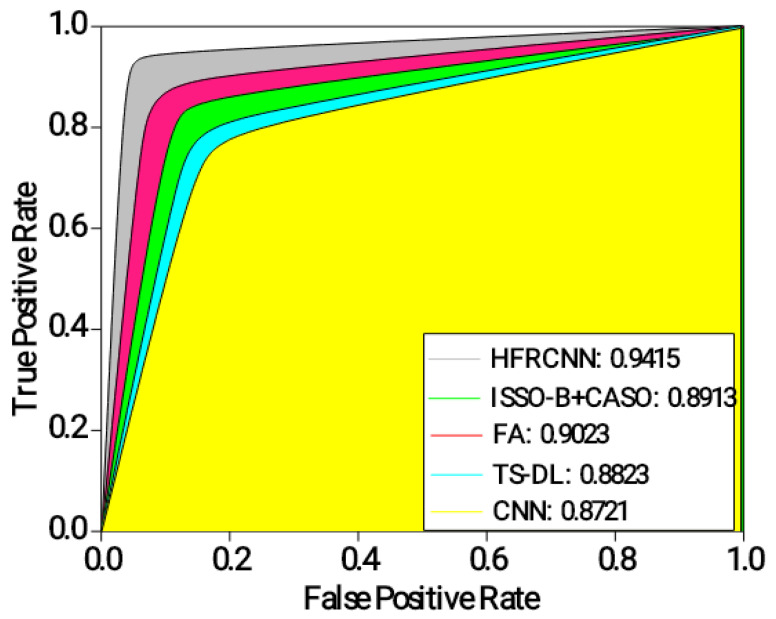
Analysis of ROC-AUC.

**Figure 6 diagnostics-13-03485-f006:**
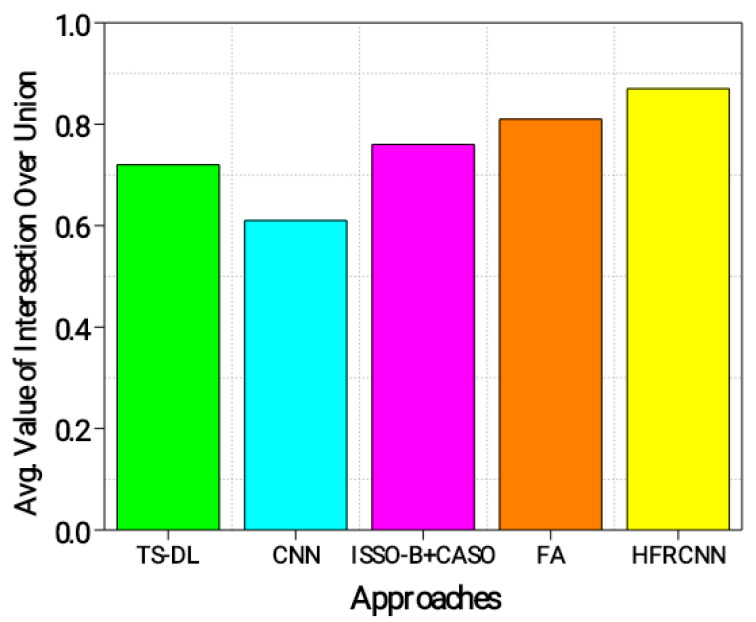
Evaluation of ǖ.

**Figure 7 diagnostics-13-03485-f007:**
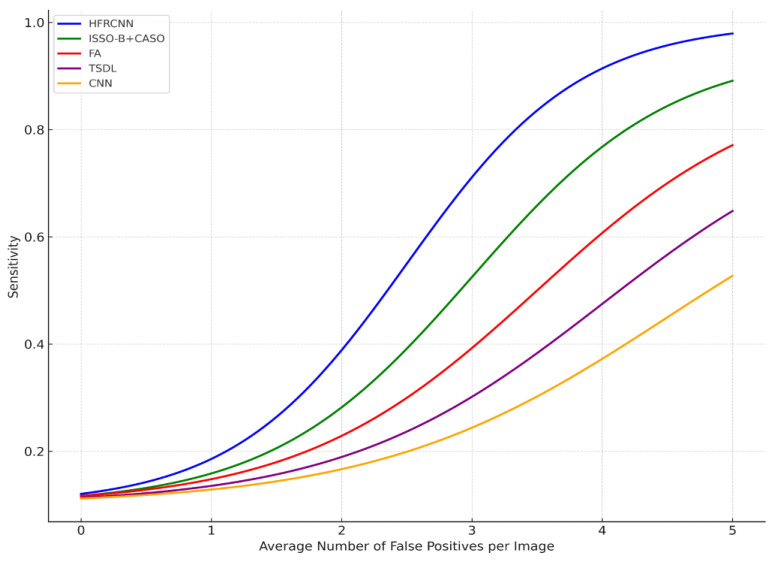
FROC curve.

**Figure 8 diagnostics-13-03485-f008:**
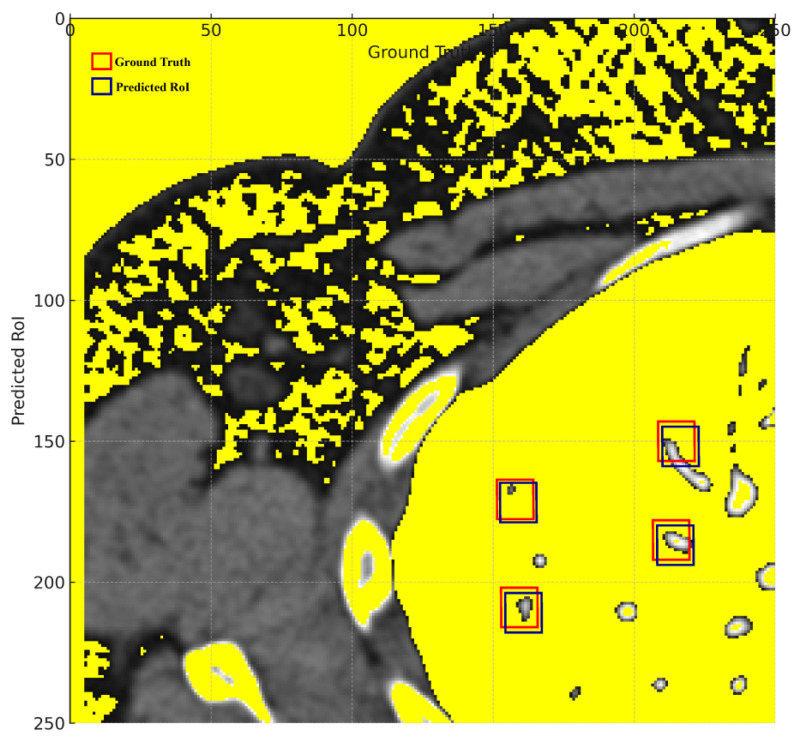
Ground truth vs. predicted ROIs.

**Figure 9 diagnostics-13-03485-f009:**
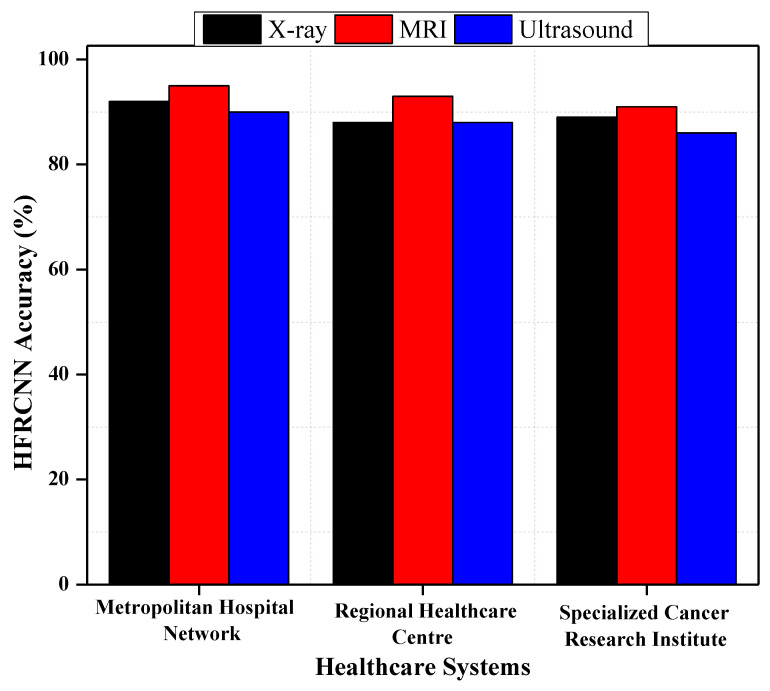
HFRCNN adaptation across different medical imaging modalities and healthcare systems.

**Figure 10 diagnostics-13-03485-f010:**
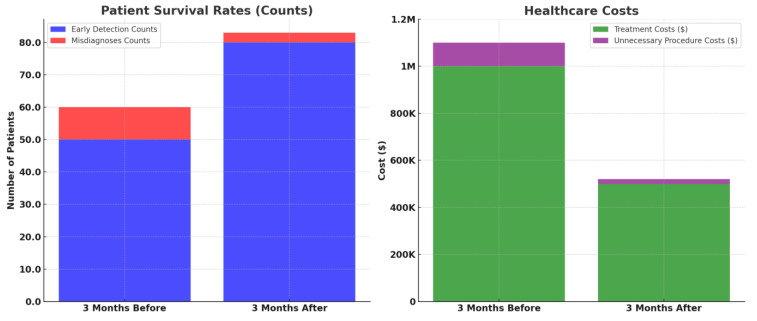
Comparative analysis of patient survival rates and healthcare costs before and after the implementation of HFRCNN over a three-month period.

**Figure 11 diagnostics-13-03485-f011:**
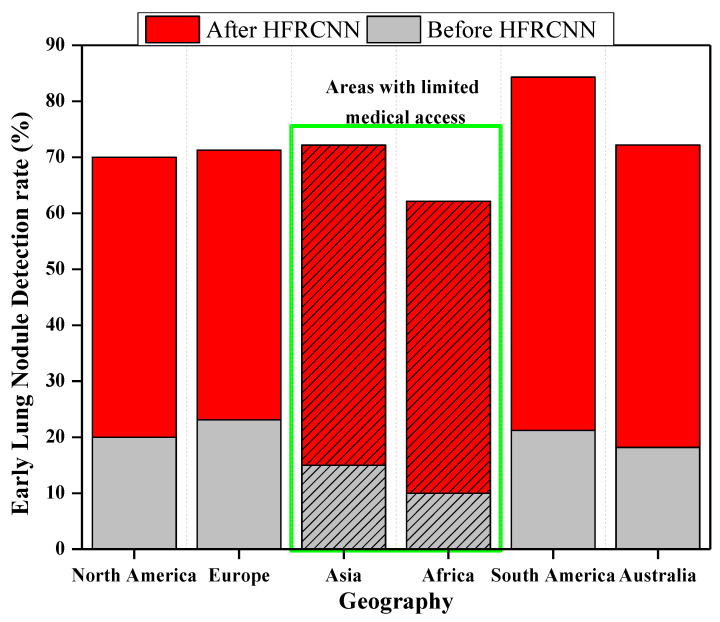
Effectiveness of the integration of HFRCNN into global telemedicine platforms.

**Table 1 diagnostics-13-03485-t001:** Consolidated review of different existing methodologies.

Study	Research Objective	Methodology	Outcomes Measured
[14]	Create an LDCT-based DL method for identifying lung nodules and analyzing their occurrence in China.	Deep learning algorithm: TS-DL	ROC (Receiver Operating Curves)–AUC (Area Under the Curve), Free-response ROC Score, Average Duration
[15]	Early identification of lung nodule anomalies using DL	U-net Design	Detection of Lung Tumor Regions, Lung Nodule Segmentation (U-Net Architecture), Lung Cancer Classification (Detecting normalcy and abnormalities)
[16]	Fast and accurate lung tumor detection (via a CNN)	CNN	Precision, Recall, ROC, AUC
[17]	Lung lesion detection and prognosis using a mixed neural network framework	ISSO-B and CASO techniques	AUC, Sensitivity, Accuracy, Specificity
[18]	Deep learning-based lung nodule detection method	Fusion Algorithms (FAs) and patch-based multi-resolution neural networks	Lung Nodule Detection, False Positives per Image (FPs/Image), FAUC (False Positive Area Under the Curve), R-CPM (Relative Cumulative Performance Measure)
[1]	Detection of malignant pulmonary nodules using deep learning from CT scans	Preprocessing pipeline to mask lung regions; feature extraction using 3D CNN based on a C3D network	Sensitivity: 86%

**Table 2 diagnostics-13-03485-t002:** Influential and characteristic features of LIDC-IDRI.

Feature	Value
Dataset Type	Medical Imaging
Dataset Size	888 CT Scans
Source	Lung Image Database Consortium (LIDC-IDRI)
Annotation Type	Expert Radiologists’ Markings for Lung Nodules
Nodule Types	Benign and Malignant
Nodule Annotations	Yes
Nodule Sizes (in millimeters, mm)	Minimum: 3 mm Maximum: 30 mm
Nodule Shapes	Round, Oval, Irregular, Spiculated, Lobulated, Spherical
Purpose	Lung Cancer Detection and Research
Released By	RSNA and NCI
Year of Release	2016

**Table 3 diagnostics-13-03485-t003:** Performance outcomes of different approaches.

Approaches	Specificity (%)	Sensitivity (%)	F1-Score (%)
TS-DL	88.16	90.14	89.14
CNN	89.23	87.34	88.32
ISSO-B + CASO	90.08	91.11	91.03
FA	92.24	90.24	91.34
HFRCNN	94.32	94.23	94.54

**Table 4 diagnostics-13-03485-t004:** Prominent factors to analyze the potential global implications of implementing HFRCNN.

Patient Survival Rates	Early detection rates Reduction in misdiagnoses
Healthcare Costs	Reduction in treatment costs due to early detection Savings from minimizing unnecessary procedures

## Data Availability

The details about the dataset are mentioned in the manuscript.

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
