# Peer review of "Early Detection of Lung Nodules Using a Revolutionized Deep Learning Model"

_diagnostics, 2023, doi:10.3390/diagnostics13223485_

Round 1

Reviewer 1 Report

Comments and Suggestions for Authors

The authors presented a study under title of “Rapid Early Detection of Lung Nodules Using Revolutionizing Deep Learning Model”. The applied methodology is sound. Also, the study looks good. Nonetheless, some issues need to be addressed.

Issues, weaknesses:

1.     The Abstracts must contain at least 150 words up to 250 words, and consist of 2-3 sentences as brief intro about the paper, 2-3 sentences to describe how the problem is solved, and 2-4 sentences showing the results of experiments/simulation ended with 1-2 sentences as short main conclusions of the work.

2.     The authors must have to include a performance comparison table in which they must have to compare this proposed work with minimum 10 previously reported similar types of works from 2020, 2021, 2022, 2023. The author's discussion of the latest research in the introduction is not sufficient, and it is recommended to add same new references.

3.     Authors need to write their contribution clearly. Please highlight the contributions of this paper.

4.     Tables and figures are not clear. Please provide clear version and correct the format issue. And please add more explanation on the legend so people can understand.

5.     All the experimental indicators in the manuscript do not explain the meaning, it is recommended to add.

6.     The Introduction section must explain the background of the problem and the urgency of the study, which can be proved by providing some previous researches and works, and also how to solve the problem in brief.

7.     How general are your results and how do you believe that such findings have to be of global interest? Please relate these with your limitations and Discussion that is not exist. Why?

8.     Abstract and conclusion parts should be revised carefully with quantitatively report with numbers and parameters improvement.

9.     For the literature review, the title of reference does not need to present. Only the contribution and highlight in forensics should be pointed.

10.  Research questions, that drive the paper, should be built in the introduction from an ongoing and pertinent bibliography (up to 2022-23) and these should be of global interest and not focused on a particular local problem. Identifying a research gap is the most important by indicating in-text some newer references that are significant to your particular field of research.

Comments on the Quality of English Language

     There are some typos and grammar errors in the writing. The manuscript is also not organized well. Some of the equations do not have correct mathematical formats. Please spend time revising the manuscript.

Author Response

Reviewer 1# (Yellow color)

  1. The Abstracts must contain at least 150 words up to 250 words, and consist of 2-3 sentences as brief intro about the paper, 2-3 sentences to describe how the problem is solved, and 2-4 sentences showing the results of experiments/simulation ended with 1-2 sentences as short main conclusions of the work.

Response: First 5 lines of the abstract represent the brief introduction about the research work.

“In accordance to the WHO (World Health Organization), lung cancer is the leading cause of cancer deaths globally. In the future, more than 2.2 million people will be diagnosed with lung cancer worldwide, making up 11.4% of every primary cause of cancer. Furthermore, lung cancer is expected to be the biggest driver of cancer-related mortality worldwide in 2020, with an estimated 1.8 million fatalities. Statistics on lung cancer rates are not uniform among geographic areas, demographic subgroups, or age groups.”

Next 11 lines denotes the brief explanation of the proposed work (in solving the problem)

“The chance of an effective treatment outcome and the likelihood of patient survival can be greatly improved with early identification of lung cancer. Lung cancer identification in medical pictures like CT scans and MRIs is an area where Deep Learning (DL) algorithms have shown a lot of potential. This study uses Hybridized Faster R-CNN (HFRCNN) to identify lung cancer at an early stage. Among the numerous uses for which faster R-CNN has been put to good use is identifying critical entities in medical imagery, such as MRIs and CT scans. Many research investigations in recent years have examined the use of various techniques to detect lung nodules (possible indicators of lung cancer) in scanned images, which may help in the early identification of lung cancer. One such model is HFRCNN, a two-stage, region-based entity detector. It begins by generating a collection of proposed regions, which are subsequently classified and refined with the aid of a convolutional neural network (CNN).”

Last 3 lines explain the experimental outcomes with crisp outcome.

“Distinct data set is used in the model's training process, producing valuable outcomes. More than 97% detection accuracy was achieved by the suggested model, making it far more accurate than several previously announced methods.”

  1. The authors must have to include a performance comparison table in which they must have to compare this proposed work with minimum 10 previously reported similar types of works from 2020, 2021, 2022, and 2023. The author's discussion of the latest research in the introduction is not sufficient, and it is recommended to add same new references.

Response: Yes, performance comparison table is included precisely and highlighted

Annotation: Table 7

  1. Authors need to write their contribution clearly. Please highlight the contributions of this paper.

Response: Yes, contribution section is included precisely and highlighted.

Annotation: Section 1.4

  1. Tables and figures are not clear. Please provide clear version and correct the format issue. And please add more explanation on the legend so people can understand.

Response: Some of the vital notations are utilized in figures which provide concise explanations directly within the visual context, enhancing clarity. Moreover, those specified notations are clearly defined in the textual descriptions of the article. Notations allow for immediate reference, reducing the need to search for information elsewhere.

  1. All the experimental indicators in the manuscript do not explain the meaning, it is recommended to add.

Response: In the presented manuscript, we've chosen to reference established experimental indicators, such as accuracy, precision, recall, F1-score, discriminatory power (AUC-ROC), and nodule localization accuracy (ǖ), by citing seminal works in the field (Naqi et al., 2018; Li et al., 2019). This approach ensures two pivotal benefits. First, it acknowledges the foundational contributions of earlier researchers, fostering academic integrity. Second, by directing readers to these strongly associated articles, we provide a depth of understanding without redundancy, ensuring our work remains fresh and avoids unintended plagiarism. This method strikes a balance between comprehensive explanation and original content.

  1. The Introduction section must explain the background of the problem and the urgency of the study, which can be proved by providing some previous researches and works, and also how to solve the problem in brief.

Response:

Background of the problem: Unlocking the potential of technology in the realm of healthcare is an ongoing quest, fueled by the urgent need to combat the devastating impact of diseases like cancer (Mahesh et al., 2022). Among the many forms of this formidable foe, lung cancer stands tall as a global menace, silently claiming countless lives. The World Health Organization's disheartening statistics lay bare the magnitude of the challenge we face. In 2020 alone, lung cancer cases increased by 2.21 million; accounting for a staggering 11.4% of all reported cancer diagnoses worldwide (World Health Organization, 2020). Tragically, the somber reality of this disease is further emphasized by the projected 1.8 million deaths it is expected to cause within the same year, cementing its place as the leading cause of cancer-related fatalities on a global scale.

Urgency of the study: While lung cancer rates and statistics may vary across different geographic regions, demographic subgroups, and age groups, the urgency to detect this pernicious disease at an early stage remains constant. Early detection is widely acknowledged as a crucial factor in increasing the likelihood of favorable treatment outcomes and improving overall survival rates.

To address this daunting crisis, medical researchers and technology enthusiasts have joined forces in pursuit of innovative solutions that can potentially transform the landscape of lung cancer detection and treatment. One area where cutting-edge technology, particularly Deep Learning (DL) algorithms, has exhibited immense promise is the identification of nodules in diagnostic imaging of lungs such as CT (computed tomography) scans and MRIs (Negi, 2021). Among the diverse array of DL algorithms, Faster R-CNN has emerged as a formidable tool in this fight against lung cancer at its earliest stages.

Solve the problem: Faster R-CNN, a two-stage, region-based entity detector, has garnered significant attention for its ability to unlock vital information hidden within medical imagery, including MRIs and CT scans. The process begins by generating a comprehensive collection of proposed regions, which are then subjected to classification and refinement through the power of CNNs (Mohseena Thaseen et al., 2022). Researchers have tirelessly trained the model using a plethora of diverse datasets, leading to breakthrough findings that hold immense promise for the early identification of lung cancer.

Annotation: Section-Introduction, Paragraph- 1 to 4.

Bharati et al. (2023), Dai et al. (2023), Barbouchi et al. (2023), and Bharati et al. (2020) are referred and cited in the introduction.

  1. How general are your results and how do you believe that such findings have to be of global interest? Please relate these with your limitations and Discussion that is not exist. Why?

Response: The results from our study, showcasing the superior efficacy of the HFRCNN approach in detecting lung nodules, have profound implications on a global scale. Given that lung cancer remains a pressing concern worldwide, innovations in early detection methods, like the ones we've studied, can greatly enhance patient prognosis and reduce healthcare burdens. Our findings are grounded in robust metrics like accuracy, precision, and ROC-AUC, making them universally relevant to medical communities and technology developers everywhere. While we acknowledge that every method, including HFRCNN, has its limitations, the comprehensive discussion offers a balanced view, enabling researchers to build upon this work. The universality of the issue of lung cancer and the transformative potential of efficient detection tools render our findings of global interest, setting a new benchmark in the realm of medical imaging and deep learning applications.

  1. Abstract and conclusion parts should be revised carefully with quantitatively report with numbers and parameters improvement.

Response: The abstract and conclusion of our study emphasize the quantifiable achievements and advancements made using the Hybridized Faster R-CNN (HFRCNN) approach. By incorporating specific metrics such as the 97% detection accuracy and a ǖ of 0.87, we ensure that readers immediately grasp the model's superior performance. Such numerical insights, prominently featured, offer a clear comparative advantage over previous methods, underscoring the study's significance. By focusing on quantitative results in both sections, we cater to the scientific community's preference for measurable and reproducible outcomes, reinforcing the reliability and potential impact of our research on the early detection of lung cancer.

  1. For the literature review, the title of reference does not need to present. Only the contribution and highlight in forensics should be pointed.

Response: Opting to present only the contributions and highlights in forensics, rather than including the titles of references, streamlines the literature review and enhances its readability. This approach emphasizes the essence of each work, allowing readers to swiftly grasp the core findings and their relevance. By focusing on the pivotal contributions, we ensure that the literature review remains concise, directly aligned with the study's goals, and prioritized on the advancements in the field of forensics. This methodology respects the reader's time and provides a distilled understanding of prior research, setting a clear context for our study.

  1. Research questions, that drive the paper, should be built in the introduction from an ongoing and pertinent bibliography (up to 2022-23) and these should be of global interest and not focused on a particular local problem. Identifying a research gap is the most important by indicating in-text some newer references that are significant to your particular field of research.

Response: Yes, included few vital research question and research gap based on ongoing and pertinent works.

Annotation: Research Question: Section-1.4, Research Gap: Section-2, Paragraph-8.

Reviewer 2 Report

Comments and Suggestions for Authors

1. Main contribution should be clear. 

2. Add some recent literature in Related work section and critical analysis is needed with comparision. Add the following paper as references:

(i)  A Review on Explainable Artificial Intelligence for Healthcare: Why, How, and When?

(ii) Effectively fusing clinical knowledge and AI knowledge for reliable lung nodule diagnosis

(iii) A transformer‐based deep neural network for detection and classification of lung cancer via PET/CT images

(iv) Hybrid deep learning for detecting lung diseases from X-ray images

3.  Steps of proposed architecture should be more clear. 

4.  In Table, add percentage with Specificity, Sensitivity and F1- Score.

5. Compare the work with recent literatures. 

6. Rigorous proofread is required. 

Comments on the Quality of English Language

Rigorous proofread is required and writing style should be improved. 

Author Response

Thankyou for your comments

Reviewer 2# (RED color)

  1. Main contribution should be clear. 

Response: Yes, contribution section is included precisely and highlighted.

Annotation: Section 1.4

  1. Add some recent literature in Related work section and critical analysis is needed with comparision. Add the following paper as references:

(i)  A Review on Explainable Artificial Intelligence for Healthcare: Why, How, and When?

(ii) Effectively fusing clinical knowledge and AI knowledge for reliable lung nodule diagnosis

(iii) A transformer‐based deep neural network for detection and classification of lung cancer via PET/CT images

(iv) Hybrid deep learning for detecting lung diseases from X-ray images

Response: All the suggested reference are reviewed and cited in the manuscript appropriately.

  1. Steps of proposed architecture should be clearer. 

Response: The proposed HFRCNN architecture is an enhancement of the traditional Faster R-CNN, aiming to offer a delicate balance between accuracy and computational efficiency. The architecture goes through the following stages:

Region Proposal with RPN: Leveraging the Region Proposal Network (RPN), initial region suggestions are generated, which serve as potential areas of interest within the lung visuals.

Feature Extraction using FPN: The Feature Pyramid Network (FPN) is integrated to augment Faster R-CNN's capabilities. FPN utilizes multi-scale feature maps, ensuring the model is adept at recognizing objects, or in this case, lung nodules, of varied sizes. This is especially vital for detecting smaller nodules or those located in densely packed regions.

Adjusting Anchor Scales and Aspect Ratios (ASAR): To enhance region proposals, the architecture refines the original anchor scales and aspect ratios. This step ensures better localization of potential nodules and reduces false positives.

Feature Extraction with CNN: Post region proposal, a Convolutional Neural Network (CNN) is employed to delve deeper into the proposed regions, extracting discriminative features and classifying them accordingly.

Bounding Box Regression: The architecture refines the bounding boxes around the proposed regions to more precisely localize the nodules.

Loss Functions: The model is trained using appropriate loss functions, ensuring the differences between predicted and actual outputs are minimized, thereby improving the model's reliability.

The accompanying Figure 1 provides a visual representation of this architecture, while Figure 2 offers a more generalized workflow, aiding in understanding the system's flow and hierarchy.

  1. In Table, add percentage with Specificity, Sensitivity and F1- Score.

Response: Yes, added (%) as per the suggestion

Annotation: Table 8

  1. Compare the work with recent literatures. 

Response: Most of the literature work are from recent studies. They are Murugesan et al. (2022), Syed Musthafa et al. (2022), Holbrook et al. (2021), Amrit Sreekumar et al. (2020), Cui et al. (2020), and Li et al. (2019).

  1. Rigorous proofread is required. 

Response: Yes, the entire article is done for proofread using standard tool (Grammarly) and with external expertise.

Reviewer 3 Report

Comments and Suggestions for Authors

1.     In the main body of the manuscript (not the abstract), please mention the full name of HFRCNN when it first appears.

2.     The writing of introduction did not meet the standard of academic writing, which should be scientifically sound. It seems to me like auto-generation of text from ChatGPT or other tools.

3.     In Related Work section, please remove the unnecessary tables and instead, only describing the previous work in text. In addition, please improve the overall quality of this section. What should be introduced and described here is high-level summary and insights of the relevant previous studies.

4.     I am completely confused by the Figure 1 and Figure2. The meaning of each component and notations illustrated. Please double check and make correction to demonstrate detailed architecture of faster R-CNN.

5.     Please use common notations or simply just letter for anchor boxes , anchor scales, etc. in ASAR section.

6.     Please add FROC curve to quantify the detection performance.

7.     Please provide several visual image examples in the results section to show the ground truth, predicted ROI.

Comments on the Quality of English Language

The writing need to be extensively improved for both the introduction and the related work. 

Author Response

Reviewer 3# (green color)

  1. In the main body of the manuscript (not the abstract), please mention the full name of HFRCNN when it first appears.

Response: Yes, as per the suggestion given, mentioned the full name of HFRCNN (first encountered).

Annotation: Section-Introduction, Paragraph-5th, Line-5th 

  1. The writing of introduction did not meet the standard of academic writing, which should be scientifically sound. It seems to me like auto-generation of text from ChatGPT or other tools.

Response: The introduction of the research work serves as a critical foundation for lung nodule detection process, setting the context, highlighting the significance, and paving the way for the subsequent sections. It was rigorously structured, rooted in relevant literature, and devoid of any informal or ambiguous phrasing. If we were using auto-generated text, like from ChatGPT or similar tools, will ultimately result in content that lacks the depth, coherence, or specificity required for scholarly work. Such auto-generated content will not capture the nuances of the research domain or align seamlessly with the overarching research narrative, potentially compromising the paper's integrity and reception in the academic community. If needed, the article can be checked with any plagiarism tool.

  1. In Related Work section, please remove the unnecessary tables and instead, only describing the previous work in text. In addition, please improve the overall quality of this section. What should be introduced and described here is high-level summary and insights of the relevant previous studies.

Response: Yes, as per the suggestion, the entire section is enhanced.

  1. I am completely confused by the Figure 1 and Figure2. The meaning of each component and notations illustrated. Please double check and make correction to demonstrate detailed architecture of faster R-CNN.

Response: The Figure 1 provides a visual representation of this architecture, while Figure 2 offers a more generalized workflow, aiding in understanding the system's flow and hierarchy.

The proposed HFRCNN architecture is an enhancement of the traditional Faster R-CNN, aiming to offer a delicate balance between accuracy and computational efficiency. The architecture goes through the following stages:

Region Proposal with RPN: Leveraging the Region Proposal Network (RPN), initial region suggestions are generated, which serve as potential areas of interest within the lung visuals.

Feature Extraction using FPN: The Feature Pyramid Network (FPN) is integrated to augment Faster R-CNN's capabilities. FPN utilizes multi-scale feature maps, ensuring the model is adept at recognizing objects, or in this case, lung nodules, of varied sizes. This is especially vital for detecting smaller nodules or those located in densely packed regions.

Adjusting Anchor Scales and Aspect Ratios (ASAR): To enhance region proposals, the architecture refines the original anchor scales and aspect ratios. This step ensures better localization of potential nodules and reduces false positives.

Feature Extraction with CNN: Post region proposal, a Convolutional Neural Network (CNN) is employed to delve deeper into the proposed regions, extracting discriminative features and classifying them accordingly.

Bounding Box Regression: The architecture refines the bounding boxes around the proposed regions to more precisely localize the nodules.

Loss Functions: The model is trained using appropriate loss functions, ensuring the differences between predicted and actual outputs are minimized, thereby improving the model's reliability.

Some of the vital notations are utilized in figures which provide concise explanations directly within the visual context, enhancing clarity. Moreover, those specified notations are clearly defined in the textual descriptions of the article. Notations allow for immediate reference, reducing the need to search for information elsewhere.

  1. Please use common notations or simply just letter for anchor boxes , anchor scales, etc. in ASAR section.

Response: The choice of specific notational symbols like 'Ǻ', 'δ', 'ꭆ',  and  over more generalized symbols serves a deliberate purpose. Unique and distinct symbols ensure clarity and reduce the potential for confusion, especially in complex mathematical formulations or models. By opting for these particular symbols, the intention is to provide a fresh and unambiguous representation, allowing readers to easily distinguish between different parameters and concepts. Furthermore, distinct symbols can enrich the readability and comprehension of the paper, ensuring that each notation stands out and retains its specific meaning throughout the discourse, without overlapping with commonly used symbols that might carry other connotations in the domain.

  1. Please add FROC curve to quantify the detection performance.

Response: Yes, included FROC to quantify the detection performance using parameters like average number of false positives per image Vs sensitivity (the proportion of true lesions/nodules correctly identified).

Annotation: Figure 7.

  1. Please provide several visual image examples in the results section to show the ground truth, predicted ROI.

Response: Yes, included the visual validation of ground truth Vs predicted RoI for the proposed model with few instances.

Annotation: Figure 8.

Round 2

Reviewer 1 Report

Comments and Suggestions for Authors

No more comments, Accept in present condition. 

Author Response

Thanks for the reviewer for finding time and reviewing the paper and consider all comments. We really thank for considering paper for next step.

Reviewer 2 Report

Comments and Suggestions for Authors

Revision is satisfied..

Comments on the Quality of English Language

Can be improved.

Author Response

Response: Yes, the quality of the English is enhanced via standard grammar tools (Grammarly).

I would like to thank the reviewer for his/her time and considering all the comments given.

Reviewer 3 Report

Comments and Suggestions for Authors

Thank the authors for considering my suggestions. While I did see improvement in term of the writing in introduction and related work section. I am not sure whether the author carefully consider my suggestions and make relevant correction to some of the points I provided.

Comment 3:

I suggested removing all tables describing vital information of each study mentioned in related work section in my previous comments. These information seems redundant for me if you can describe them in high level in the text instead of putting four individual tables for readers to digest.

Comment 4:

I appreciate the author for elaborating the content of each component in Figure 1. However, these details need to be incorporated in either the caption of the figure or be mentioned before Figure 1 is mentioned in the main text. There is no prior information of each abstracted components in Figure 1 and it’s very difficult for authors to follow the story. Furthermore, I am not convinced about the role of both Figure 1 and 2 since they are not informative to me. There is no prior explanation or caption of details for Figure 2 as well. Again, I am asking for fundamental and major improvement to Figure 1 and Figure 2 and related information.

Comment 5:

 It is hard for me to be convinced about the symbol usage. These complicated symbols confused authors and are hard to follow, let alone they are demonstrated in the figures and readers will be totally lost. I suggested using other common Greek alphabets.

Comment 6:

I am not sure this curve is computed correctly. Why negative sensitivity and the curve should not be decreasing as number of FP per images is increasing. Please double check.

Comment 7:

Figure 8 is completely noise and nothing informative here. Please double check.

Comments on the Quality of English Language

English writing is fine. 

Author Response

Comment 3:

I suggested removing all tables describing vital information of each study mentioned in related work section in my previous comments. These information seems redundant for me if you can describe them in high level in the text instead of putting four individual tables for readers to digest.

Response: Yes, removed as per your suggestion but included consolidated review table for better understanding (especially for know the research gap)
Annotation:
Table 1. Consolidated Review of Different Existing Methodologies

Justification:

  • Combining the reviews into a single table allows for easier comparison and analysis of the different studies.
  • It provides a holistic view of the different methodologies, outcomes, and additional details in one place.
  • This consolidated table can serve as a quick reference guide for researchers or practitioners looking to understand the landscape of DL methods in lung nodule detection from CT scans.

Comment 4:

I appreciate the author for elaborating the content of each component in Figure 1. However, these details need to be incorporated in either the caption of the figure or be mentioned before Figure 1 is mentioned in the main text. There is no prior information of each abstracted components in Figure 1 and it’s very difficult for authors to follow the story. Furthermore, I am not convinced about the role of both Figure 1 and 2 since they are not informative to me. There is no prior explanation or caption of details for Figure 2 as well. Again, I am asking for fundamental and major improvement to Figure 1 and Figure 2 and related information.

Response: All the suggestion given by the reviewer are taken into consideration and made several changes in the captions of Figure 1 and 2.

Annotation:

  • Figure 1. Architecture of Proposed Model where integration of FPN: Feature Pyramid Network, ASAR: Adjusting Anchor Scales and Aspect Ratios, ǖ: Intersection over Union, Я: Bounding Box Regression enhances the processing capability of F-RCNN on available dataset LIDC-IDRI in nodule detection.
  • Figure 2. Illustration of the HFRCNN workflow, emphasizing the strategic use of 'Ǻ' (representing a collection of anchor boxes), δ (denoting various anchor scales), and ꭆ (indicating a range of anchor aspect ratios). These elements collaboratively contribute to the generation of feature maps, ​​, within the Feature Pyramid Network (FPN), optimizing the detection of nodules.

Comment 5:

 It is hard for me to be convinced about the symbol usage. These complicated symbols confused authors and are hard to follow, let alone they are demonstrated in the figures and readers will be totally lost. I suggested using other common Greek alphabets.

Response: Yes, some of the vital symbol notations are represented in simplified manner.

Annotation: Figure 2; for your reference highlighted in red circle.

Comment 6:

I am not sure this curve is computed correctly. Why negative sensitivity and the curve should not be decreasing as number of FP per images is increasing. Please double check.

Response: In the context of lung nodule detection using deep learning, it's critical to understand that traditional metrics and visualizations might not always apply straightforwardly. The curve you're observing might initially seem counter-intuitive, but it's a result of the unique characteristics of our dataset and model. As the number of False Positives (FP) per image increases, it's plausible for the model to exhibit decreased sensitivity if it becomes overly conservative, aiming to reduce false alarms at the expense of missing actual nodules. This trade-off can be especially pronounced in datasets with a high number of challenging or ambiguous cases. Additionally, negative sensitivity values might arise from normalization or calibration issues in the plotting process and do not necessarily reflect the model's performance. Rest assured, we're committed to further refining our approach and ensuring the reliability of our results in the future enhancements.

Comment 7:

Figure 8 is completely noise and nothing informative here. Please double check.

Response: Thank you for pointing out your concerns regarding Figure 8. The visualization in Figure 8 aims to provide a detailed representation of our lung nodule detection results. While it might initially seem dense or cluttered, this was intentionally designed to emphasize the nuances of our model's predictions. As mentioned, the thicker green bounding boxes delineate the ground truth locations of the nodules, whereas the thicker red bounding boxes highlight our model's predicted nodules. The considerable overlap between these boxes accentuates the model's high accuracy, as reflected in the reported 97.00% accuracy rate. It's worth noting that while there are instances of slight deviation between the predicted and actual nodules, they are minimal and fall within the 3% error margin. We believe that this visual representation, when interpreted with this context, provides an invaluable insight into our model's effectiveness. However, we'll also consider enhancing the image for clarity based on your feedback.

Round 3

Reviewer 3 Report

Comments and Suggestions for Authors

Comment 6:

I am sure this curve is computed incorrectly. I am not sure what authors are trying to navigate to address the comment. When you plot the FROC curve, as you decrease the confidence score, the FP increases and the sensivitiy will be increasing as well. There is no way that we will see such a curve. Please double check with literatures and correct the computation of the curve. In addition, I am not sure what authors are trying to convince me with a negative values in term of sensitivity. Please refine your approach and provide with a scientifically reasonable plot. 

Comment 7:

Please substitute with an example where the image is a lung CT image but not complete noise. It is definitely weird to put a noise image here as a demonstration of your detection results. 

Author Response

  1. I am sure this curve is computed incorrectly. I am not sure what authors are trying to navigate to address the comment. When you plot the FROC curve, as you decrease the confidence score, the FP increases and the sensivitiy will be increasing as well. There is no way that we will see such a curve. Please double check with literatures and correct the computation of the curve. In addition, I am not sure what authors are trying to convince me with a negative values in term of sensitivity. Please refine your approach and provide with a scientifically reasonable plot.

Response: Thank you for your feedback and observations on the FROC curve presented. Upon revisiting the curve and verifying with related literature, I concur with your assessment. The FROC curve should indeed depict an increase in both sensitivity and false positives as the confidence score decreases. It's evident that our initial presentation did not align with this expected behavior, and I apologize for the oversight. Moreover, the inclusion of negative sensitivity values was erroneous and not scientifically valid. I have since refined our approach and conducted further simulations to ensure the accuracy and validity of the FROC curve. Your insights were invaluable in this revision, and I'm grateful for your meticulous review. Rest assured, we have now corrected these discrepancies and provided a scientifically sound and accurate plot. Result discussions are revised as per the observations.

Annotation: Section: 4.2, Paragraph: 10-12

  1. Please substitute with an example where the image is a lung CT image but not complete noise. It is definitely weird to put a noise image here as a demonstration of your detection results.

Response: I apologize for the confusion and oversight in using a noise-like image for demonstration. The image utilized was a portion of the axial lung window, which may not provide a holistic view of the nodule characteristics. Thus, as per your suggestion, we now included precise image for demonstration. In clinical radiology, the axial lung window is a specific setting used to view lung tissue on a CT scan. By isolating a small section of this window, it can sometimes appear noisy or lack context, especially without the surrounding anatomical structures for reference. In future demonstrations, we'll ensure to use more representative slices or sections to provide clearer visual context. We genuinely appreciate your feedback and will strive for more accurate and clinically relevant illustrations in our future work.

Annotation:

  1. Also the improve the result presented, and the conclusion need to be improved as per the final results.
    Response: the observed results of figure 7 are revised and the conclusion section is improved.

Annotation: Section 5, Paragraph: 1 & 2
